# *Return to 1616*: Multispecies Fauna Reconstruction Requires Thinking Outside the Box

**DOI:** 10.3390/ani13172762

**Published:** 2023-08-30

**Authors:** Saul Cowen, Colleen Sims, Kym Ottewell, Fiona Knox, Tony Friend, Harriet Mills, Sean Garretson, Kelly Rayner, Lesley Gibson

**Affiliations:** 1Biodiversity and Conservation Science, Department of Biodiversity, Conservation and Attractions, Woodvale, WA 6026, Australia; colleen.sims@dbca.wa.gov.au (C.S.); fiona.knox@murdoch.edu.au (F.K.); sean.garretson@dbca.wa.gov.au (S.G.); kelly.rayner@dbca.wa.gov.au (K.R.); lesley.gibson@dbca.wa.gov.au (L.G.); 2School of Biological Sciences, University of Western Australia, Crawley, WA 6009, Australia; 3Biodiversity and Conservation Science, Department of Biodiversity, Conservation and Attractions, Kensington, WA 6151, Australia; kym.ottewell@dbca.wa.gov.au; 4School of Veterinary Medicine, Murdoch University, Murdoch, WA 6150, Australia; 5Biodiversity and Conservation Science, Department of Biodiversity, Conservation and Attractions, Albany, WA 6330, Australia; tony.friend@dbca.wa.gov.au; 6Biodiversity and Conservation Science, Department of Biodiversity, Conservation and Attractions, South Perth, WA 6951, Australia; harriet.mills@dbca.wa.gov.au

**Keywords:** conservation translocation, ecological restoration, wildlife monitoring, wildlife disease, conservation genetics, population modelling, animal welfare

## Abstract

**Simple Summary:**

Efforts are underway to restore the largest island in Western Australia to an ecological state similar to how it was when Europeans first visited the island in 1616. This has entailed the reintroduction of many species of animals, presumed to have been driven to extinction on the island due to non-native fauna species, which have now been removed. Such a large and complex project required considerable pre-planning and innovative thinking. We used a variety of techniques to assist with decision making and the implementation of this project.

**Abstract:**

Conservation translocations have become increasingly popular for ‘rewilding’ areas that have lost their native fauna. These multispecies translocations are complex and need to consider the requirements of each individual species as well as the influence of likely interactions among them. The Dirk Hartog Island National Park Ecological Restoration Project, *Return to 1616*, aspires to restore ecological function to Western Australia’s largest island. Since 2012, pest animals have been eradicated, and conservation translocations of seven fauna species have been undertaken, with a further six planned. Here, we present a synthesis of the innovative approaches undertaken in restoring the former faunal assemblage of Dirk Hartog Island and the key learnings gathered as the project has progressed.

## 1. Introduction

Conservation translocations (hereafter ‘translocations’) are the intentional movement of organisms to enact a net conservation benefit [1,2]. Translocations may be undertaken for a variety of reasons, such as population restoration via reinforcement or reintroduction or establishing new populations outside of their former indigenous range (assisted colonisation or ecological replacement). Globally, translocations are a widely used management tool for a range of animal taxa [3,4,5], and best practice guidelines have been developed that advise key considerations for effective and ethical translocation [2]. Invasive predators are a key driver of global biodiversity loss, not least in Australia [6], and mitigating this is an important motivation for undertaking translocations [1]. In Australia, translocations of threatened fauna to ‘safe havens’ (i.e., fenced reserves or islands free from non-native predators) have been highly effective management actions [7]. However, there is substantial value in the publication and dissemination of case studies to provide context for managers of ongoing and future translocations [8]. The success of translocations varies considerably [3,5,9], and it is vital that practitioners and researchers continue to share the outcomes and learnings from their efforts. In Australia, translocations of animals often involve the movement of multiple species to a single site (e.g., [10,11,12,13,14]). These ‘multispecies’ translocations naturally impose greater complexity and logistical challenges than single-species translocations, especially given the interactions between translocated species, which also need to be considered [1].

The Dirk Hartog Island National Park Ecological Restoration Project or *Return to 1616* seeks to restore Dirk Hartog Island (DHI) to a similar functional ecological state to that which existed when the first Europeans visited the island in 1616. The project is managed by the Western Australian Department of Biodiversity, Conservation and Attractions (DBCA). At 63,300 ha (~80 km long and 3–11 km wide), the island is Western Australia’s largest. When the project commenced in 2012, it set the targets of eradicating sheep (*Ovis aries*), goats (*Capra hircus*) and feral cats (*Felis catus*) and undertaking a multispecies translocation program to restore the locally extinct faunal assemblage. Thirteen species of fauna, at least eleven of which were known to have been previously present on DHI [15], were selected for translocation [10]. With planned mammal eradications completed by 2018, DHI became the largest island in the world where goats and feral cats have each been removed [16,17]. The management of non-native plants on the island is ongoing [18], with the targets of eradicating high priority species whilst avoiding the introduction of new species or the spread of existing ones [15].

Dirk Hartog Island is situated in the Shark Bay World Heritage Area, on the west coast of Western Australia (WA) (−25.8° S, −113.1° E; Figure 1) and in the country of the Malgana people, to whom it is known as *Wirruwana*. The first recorded Europeans to visit the island were on board the Dutch ship *Eendracht*, captained by Dirk Hartog for whom the island was later named. The island was used to graze sheep from the 1860s to 2009, and goats became established in the early 20th century. Feral cats are believed to have become established in the late 19th century [19]. Sheep and goats caused considerable damage to the island’s vegetation, especially around artificial sources of water. The overall impact of ungulates on the island’s vegetation can best be understood by the rapid and significant recovery since the eradications [20]. However, it was the combined impact of sheep, goats and feral cats that is presumed to have led to the extinction of at least three-quarters of the native mammal fauna [21], with just three non-volant native species remaining extant by the commencement of *Return to 1616*.

Two species of non-native vertebrates remain established on the island: house mouse (*Mus musculus*) and laughing dove (*Spilopelia senegalensis*). Laughing doves are not considered to be an invasive species in Australia, and their presence was not viewed as a threat to the extant or translocated fauna. House mice are an invasive species that may outcompete native species [22], but, based on estimates for another WA island [23], the eradication of house mice from an area the size of DHI would cost more than USD 47 million. Furthermore, any eradication program would likely endanger the extant (and translocated) small vertebrates on the island. The population dynamics of house mice are monitored annually using trapping surveys (see Section 2). Monitoring indicates that native rodents are more abundant than house mice [24], which are not currently considered to be a problem. European rabbits (*Oryctolagus cuniculus*), rats (*Rattus* spp.) and red foxes (*Vulpes vulpes*) are important invasive species across Australia but have never established populations on DHI.

The restoration of the island’s former faunal assemblage (‘fauna reconstruction’; see Appendix A) commenced in 2017 with a trial translocation of rufous hare-wallabies (*Lagorchestes hirsutus*) and banded hare-wallabies (*Lagostrophus fasciatus*), involving 12 individuals of each species [25]. The success of this trial (with just one recorded mortality out of 24 animals after nine months [26]) paved the way for full-scale translocations of these species, followed by five more mammals and one bird (Table 1). Five more species have been proposed for reintroduction to the island (Table 1) [10,19]. Of these 13 species, 10 are listed as threatened under the Australian Commonwealth Environmental Protection and Biodiversity Conservation (EPBC) Act 1999. Translocations are expected to continue until 2027, with funding for *Return to 1616* to cease by 30 June 2030. Ten of the twelve mammal species planned for translocation have been identified as having previously occurred on the island from subfossil remains [27,28]. Due to the absence of any material evidence, the two hare-wallaby species are not definitively known to have occurred on DHI [21,27]. However, there are natural populations on Bernier Island and Dorre Island nearby, and there is compelling anecdotal evidence (e.g., [29,30]) that they did occur naturally on DHI. Specimens of western grasswren were collected from the island in the 20th century [31], but the species is believed to have become locally extinct around 1920 [32]. Two other species are thought to have become extinct on DHI: the rock parrot (*Neophema petrophila*), which was previously observed on the island but is now apparently extinct [33], and the rakali or water rat (*Hydromys chrysogaster*), which was speculated to have also occurred [10]. However, both species were considered capable of natural recolonisation from nearby populations in Shark Bay. No other taxa, including plants, amphibians, reptiles or invertebrates, are known to have been extirpated from DHI.

The apex terrestrial predator on DHI was formerly the chuditch or western quoll (*Dasyurus geoffroii*), but in its absence, there are other predators of small vertebrates, such as the sand monitor (*Varanus gouldii*), mulga snake (*Pseudechis australis*), gwardar (or western brown snake) (*Pseudonaja mengdeni*), Children’s python (*Antaresia childreni*) and grey butcherbird (*Cracticus torquatus*). The two largest predators found on the island are the wedge-tailed eagle (*Aquila audax*) and the white-bellied sea eagle (*Haliaeetus leucogaster*), with the former being an apparently scarce visitor to DHI. All these species are natural predators of the former fauna assemblage of the island.

This number of species to be translocated over such a large landscape represents one of the more ambitious fauna reconstruction programs in Australia and globally. Most of these species are restricted to small populations, often on islands, where they may be vulnerable to the introduction of disease or non-native predators, fire, increased impacts of drought and climate change, or even overharvesting for translocation. Translocation programs in Australia have historically had an even likelihood of failure [34,35] and have often been poorly monitored [36,37]. Ethically and legally, translocation projects must give the utmost consideration to the conservation status and individual welfare of the animals. Achieving a successful outcome for the fauna reconstruction element of *Return to 1616* presents an enormous challenge. Here, we review the planning, decision making and adaptive processes that have been undertaken so far to carefully balance the positive aims of the project with the inherent risks and uncertainty that translocation programs face and the demands of upholding the legislative obligations surrounding threatened species and animal welfare.

## 2. Interspecific Interactions

In common with many other multispecies translocation programs, the DHI fauna reconstruction seeks to restore the island’s ecosystems to a similar condition to their pre-European state, including all the various interspecific interactions between predators and prey, as well as competition and cooperation. This inherent complexity is compounded by the dynamic effect of introducing relatively small numbers of founders of species (compared to the overall carrying capacity of a 63,300 ha island) that are likely to have different population growth rates. Many of the species proposed for translocation have not co-occurred in the modern research era, meaning much of the planning around species interactions is educated guesswork. A strategic framework for the DHI fauna reconstruction was produced prior to the commencement of the translocations [10], although the need to be flexible was recognised given practical considerations (such as the availability of viable source populations) and logistical constraints. There was also some concern that potential negative interspecific interactions may need to be mitigated by varying the relative timing and location of some translocations.

To address this high level of uncertainty, a risk analysis was undertaken using ensemble modelling to compare 23 different possible translocation scenarios [38]. Scenarios varied in the timing, order and location of species translocations, with the aim of identifying optimal strategies to avoid negative translocation outcomes. A priori interactions that were identified as being potentially problematic for the program’s success were competition between boodies and other non-carnivorous species; predation by brush-tailed mulgara and chuditch; and competition between desert mice and other rodents, especially heath mice. The scenarios included strategies to potentially mitigate these interactions. However, the models identified other species-level interactions that had the highest probability of leading to translocation failure for certain species. For example, for the translocation of dibblers, competition with Shark Bay bandicoots was predicted to be more important in the outcome of the translocation than interactions with extant predators (e.g., predatory birds and reptiles) and competitors. The most important outcome of the modelling was that, for all scenarios, most (if not all) species were predicted to successfully establish on the island. This provided confidence that varying the order of translocations, either to minimise negative interactions or for logistical reasons, was unlikely to have a significant negative impact on the overall outcome of the project. It also gave the project greater transparency around the reasons and support for varying the original status quo strategy and demonstrated the value in undertaking modelling exercises such as this for large multispecies translocations.

Another unpredictable effect of translocations is the potential impact on extant fauna assemblages. DHI also has the highest diversity in herpetofauna of any WA island, with over 40 species recorded [39], including some threatened and priority species. There are also three native terrestrial mammals that have persisted on the island: the ash-grey mouse (*Pseudomys albocinereus*), sandy inland mouse (*Pseudomys hermannsburgensis*) and little long-tailed dunnart (*Sminthopsis dolichura*). While extant small vertebrates on DHI were included in the model ensembles above, an ongoing monitoring program was also initiated to observe population trends, not just in response to translocations but also in response to the eradications of feral cats, goats and sheep. This program uses a combination of pitfall traps and metal box traps to detect a broad range of small vertebrates. A monitoring baseline was established even before DHI became a national park in 2009, with six surveys taking place at eight sites between 2005 and 2011. The program was revived in 2017 prior to any small mammal (<350 g) reintroductions and is ongoing. Surveys now take place in mid-spring (mid-to-late October) each year for seven nights. The abundance of small vertebrates is likely to be dynamic in response to climatic conditions (both during and prior to the survey), and to elucidate any trends that may relate to the eradications and subsequent translocations, it is important that long-term data continue to be collected. To date, six years of data have been collected annually since 2017, with plans to analyse these data following the 2023 survey.

## 3. Wild-to-Wild vs. Captive Breeding

When considering the source of founders for translocation, programs may choose to harvest from wild populations or use progeny from captive-breeding programs, depending on availability. Captive-breeding is a popular strategy [40], but the success rate of translocations using captive-sourced founders is apparently lower than that using wild-sourced translocations [37,41] for a variety of reasons [9]. Some of the challenges that translocations of captive-bred founders may face are low genetic diversity and inbreeding [42,43], loss of key behaviours [9,44,45] and disease [46,47]. However, translocations that use captive-bred founders tend to involve more animals and may also assist with safeguarding wild populations [48], although there does appear to be some variation in outcomes between different taxa, with captive-bred numbats (*Myrmecobius fasciatus*) performing no differently to wild-born animals [49]. Nevertheless, adequately addressing these challenges is essential from an animal welfare viewpoint, as well as for the overall success of the translocation [40].

Of the eight species that have been translocated (Table 1), seven have previously been bred successfully in captivity [50,51,52,53,54]. Only western grasswren has not been bred in captivity. Therefore, it was theoretically feasible that translocations to DHI could be sourced using captive-bred founders, which would reduce pressure on the wild populations. However, the cost of establishing and maintaining six captive-breeding programs at the scale required to achieve the establishment of large and genetically representative populations would have required the allocation of resources beyond those available to the project. The advantages of sourcing wild animals for translocations (i.e., adequate genetic diversity and maintenance of natural behaviours) were considered to balance the risk of detrimental impact to the wild populations. However, to better understand and manage this risk, the pre- and post-harvest monitoring of source populations was undertaken (see Section 4).

For the *Return to 1616* project, one species was identified as being suitable for captive-breeding, the dibbler. When considering source populations for translocation to DHI, the three island populations off Jurien Bay (Figure 1; Table 1) were identified as being the most appropriate. Dibblers occur naturally on two Jurien Bay islands (Boullanger and Whitlock), and a third translocated population exists on Escape Island. The latter was established from the captive-bred progeny of a mix of Boullanger and Whitlock founders from 1998 to 2000. It was hypothesised that dibblers on these islands would be better adapted to the climatic and insular conditions on DHI, and there was value in establishing a new population of this genetically divergent lineage [55] in another location (D. Moro and T. Friend *unpublished* [56]). Populations of dibblers on the three Jurien Bay islands have been historically small (occasionally < 30 on some islands) [57], and directly harvesting founders for a viable wild-to-wild translocation cohort for DHI would likely be detrimental. As such, small numbers from each island were brought into captivity at Perth Zoo to establish a maximum of 10 breeding pairs per breeding season. Dibblers are monoestrous, breeding once per year in autumn, but females may produce up to eight young per litter. This theoretically gave the breeding program the capacity to produce up to 80 young per year for release on DHI. As dibblers have been bred at Perth Zoo since 1997 and husbandry protocols for the species have been well established [52], a total of 150 individuals to be released on DHI within three years was considered to be a realistic goal (D. Moro and T. Friend *unpublished*).

Between November 2018 and February 2019, 18 individuals (15 sub-adults) were sourced from Whitlock (*n* = 9) and Escape Islands (*n* = 9). One sub-adult from each island died under general anaesthetic during routine health-checks at Perth Zoo, and they were subsequently replaced. Eight pairs were available for breeding in 2019. Three males from Boullanger Island were added in October 2019, as were three pairs from Escape Island in December 2020 and three more pairs from Escape Island in December 2022. The breeding of captive dibblers at Perth Zoo has generally been successful, but output was lower than expected in 2019 and 2020, with 50% of females failing to breed. Consequently, just 26 (24 offspring) and 31 individuals (17 offspring) were released on DHI in 2019 and 2020, respectively. It appeared that three females either failed to enter oestrus or entered oestrus before they were paired with males, but the underlying cause is still unknown. Experimental trials in the monoestrous and closely related agile antechinus (*Antechinus agilis*) found that females offered multiple males chose to mate more with males that were more genetically dissimilar, and these males sired substantially more offspring [58]. Therefore, in 2021, female dibblers were given access to two males instead of just one. This resulted in 70% of females producing young, and 36 dibblers (28 offspring) were released on DHI. In 2022, this method was augmented further to alternate access to one of two males every 24–72 h [56]. In 2022, the number of offspring released on DHI increased to 33 (of a total of 44). By 2022, a total of 137 dibblers had been released in four years, which was still short of the target of 150 over three years. Of these, 25% were adults presumed to have reached the end of their reproductive lives.

Aside from the lower than predicted number of progeny from the captive-breeding program, there was another unforeseen issue with using founders from a captive source that affected the project’s ability to monitor the short-term outcomes of the translocation. Radio tracking dibblers in 2019 and 2020 employed the use of collars to monitor early rates of survival and movement in the release area. Thirteen collars were deployed, of which at least nine were functional during the first two weeks post-release. Seven of these either slipped off or became entangled in limbs or the animal’s body. After this was observed in 2019, collar trials were conducted on captive animals at Perth Zoo (C. Sims et al. *unpublished*) to improve the collar fit. However, despite optimising the fit of collars under general anaesthetic and monitoring collar wear in captivity, it appeared that initial weight loss after the translocation resulted in collars becoming loose, allowing animals to entrap their forelimbs or shed their collars entirely. Animals sourced from captivity may be more prone to weight loss during translocation [59,60], and even captive-bred dibblers initially held in soft-release pens (with food provided) exhibited weight loss [24]. In contrast, a radio-tracking study of wild dibblers did not encounter this issue [61]. Consequently, the use of radio-collars on captive-bred dibblers released on DHI was suspended for welfare reasons, and alternative methods to assess the progress towards meeting success criteria for the translocation were identified (see Section 5). However, this highlights an unforeseen issue with sourcing founders from captivity that other programs may need to consider.

## 4. Source Population Viability

A critical consideration of any wild-to-wild translocation is the risk to the source population being harvested [2], something which is not commonly considered [62]. While the benefit of translocations to establish new populations (and thereby reduce overall extinction risk to a species) may seem implicit, this benefit only exists if the translocation is ultimately successful, and the source population(s) are not significantly affected. Preferred source populations are often remnant natural populations, in which resides the remaining genetic diversity (ergo adaptive potential) of the species. However, these populations are also the most valuable, and therefore it is critical that they are not detrimentally affected by harvesting for translocation.

Ensuring that translocations do not risk the viability of source populations usually requires adequate demographic data shortly before a translocation, obtained using a robustly designed survey. However, for some populations, obtaining robust datasets may prove challenging. For example, monitoring source populations of banded and rufous hare-wallabies on Bernier and Dorre Islands involved distance sampling surveys using spotlights at night [63]. Distance sampling can be a robust method for estimating population density [64], if its assumptions hold. However, the method relies on obtaining >60 detections to obtain a reliable estimate [65], which may not be possible with cryptic species such as banded hare-wallabies [66]. As such, a heuristic approach was taken for estimates of the hare-wallaby species, using the lower confidence limit for the population estimate from each island and ensuring that no more than 10% of this value was harvested in any one year. This approach aimed to ensure that the harvest from each island was unlikely to cause a measurable adverse impact on the population. Pre- and post-source population monitoring was undertaken for each species translocated to DHI, and there has been no evidence of detrimental impacts from translocations in any of these cases [63,67,68,69,70]. The exception was the greater stick-nest rat translocation from the Franklin Islands in South Australia, where no post-harvest monitoring was undertaken, in part due to the logistics and costs involved. In this case, a heuristic method was combined with a modelling approach (see below) to minimise the risk of the detrimental impact on this critically important population.

Population viability analyses can be useful decision making tools, incorporating a species’ biological and demographic parameters to make predictions about how a population’s trajectory may change under different scenarios [71,72]. Population viability analyses (PVA) were undertaken for several species that were translocated to DHI, namely banded hare-wallaby [73], dibbler [57], Shark Bay bandicoot [74], Shark Bay mouse [75], greater stick-nest rat [76] and western grasswren (A. Gibson Vega et al. *unpublished*). In each case, the impact on both the source population(s) and the translocated population was considered in the PVA. Genetic data were also incorporated to model how different harvesting scenarios influenced how genetically diverse the resulting translocated population was likely to be, as well as the impact to the genetic diversity of the source population(s). The greater stick-nest rat example is useful in demonstrating the value of PVA in understanding the preservation of source populations. The primary aim was to optimise the number of founders released on DHI from the natural population on the Franklin Islands (where much of the available genetic diversity resides) and the translocated population on Salutation Island (which is genetically depauperate but more proximal to DHI [77] (Figure 1)). The PVA suggested that 60 individuals from each population was sufficient to successfully establish a genetically healthy and representative population on DHI, whilst having a negligible impact on either source population. However, similar results could be achieved if the founder cohort was skewed towards the more genetically depauperate Salutation Island population, therefore reducing the number of animals required from South Australia. This demonstrates the value in using PVA to optimise translocation strategies whilst still protecting important source populations.

## 5. Release Strategies

Strategies for the release of animals require careful consideration as they are likely to influence the success of a translocation. Release strategies can be ‘hard’ (i.e., no provision of resources) or ‘soft’ (also called ‘delayed’) where resources such as food and shelter are provided during a temporary period of captivity at the release site. Timing is another important consideration. Several species that have been translocated to DHI are at risk to predation by extant reptiles, which are more active during the warmer months (October to March). A review of the outcomes of translocations of greater stick-nest rats found that, in areas where monitor lizards (*Varanus* spp.) are present, translocations that took place in autumn or winter were more likely to be successful [53]. Therefore, where feasible, we elected to release those species most vulnerable to reptile predation (e.g., greater stick-nest rat and Shark Bay mouse) in the cooler months of April to July.

A key consideration when planning a release is the origin of the founders [78]. For wild-sourced founders, a hard release strategy may be more appropriate due to the additional stress that confinement during a soft release may impose. In contrast, captive-bred founders may adapt more readily to a soft release strategy [79]. However, this is not always the case as responses are likely to be dependent on specific behavioural traits [80]. Another option is to undertake a hard release (i.e., without a period of confinement) but provide resources such as shelter which can assist with acclimatisation during the critical early stages of a translocation.

Release strategy may also be important for increasing monitoring efficacy and, hence, adequately measuring translocation success [81,82,83]. For example, soft releases have been demonstrated to improve site fidelity (i.e., reducing dispersal distance from the release area) in some translocations [80,84,85,86], which is likely to assist monitoring efforts. As six out of the seven species translocated to DHI to date were sourced from wild populations (Table 1), soft releases were not considered for these species. The two hare-wallaby species and Shark Bay bandicoots were monitored intensively using radio tracking to assess short-term monitoring success. The high rate of survival of these species and the relatively short distances dispersed [25,26,87,88] validated this decision. However, the post-release monitoring of Shark Bay mice recently translocated to another site has proven challenging (Australian Wildlife Conservancy *unpublished data*). Previous experience also suggests that translocated greater stick-nest rats can exhibit ‘hyperdispersal’ [89], which may compromise both fitness and monitoring efficacy [53]. Consequently, the provision of artificial refuges was employed in this project to improve monitoring efficacy and reduce dispersal.

Shark Bay mice on Bernier Island construct and inhabit shallow but complex burrow systems with multiple chambers (P. Speldewinde *unpublished data*). Therefore, simplified versions of these burrow systems, constructed from plastic pipes buried in soft sand, were provided as an initial refuge for recently released animals. During the 2021 release, two such artificial refuges were placed at each of 10 release sites, with eight mice released at each site. Refuges were coined ‘pseud-home-ys’ sites in reference to the genus. Camera traps were placed on one pseud-home-ys at each site to monitor activity. Shark Bay mice were recorded at all 10 pseud-home-ys sites for the duration of the 11-month deployment of the cameras and were regularly seen entering the refuges and interacting with conspecifics in their vicinity. Shark Bay mice are relatively large compared to other small rodents on DHI, but this difference may be hard to discern on camera trap images. The aperture of the pseud-home-ys also provided a reference point by which different rodent species could be discriminated [24].

As their name suggests, greater stick-nest rats construct above-ground nests consisting of woody material, which were amalgamated using their unusually viscous urine [90]. These nests provide refuge from predators and thermal extremes [53,91]. In this case, we provided artificial nests (or ‘proto-nests’) designed to mimic the structure of their natural nests. Animals were released from specially designed wooden transport boxes, directly into proto-nests after sunset and allowed to leave in their own time with disturbance minimised as much as possible. Boxes were left in situ as an additional refuge. The translocation was timed for late autumn (late May) to coincide with the period of the lowest activity of predatory reptiles such as sand monitors. In 2021, presumed social groups that had been captured from the vicinity of the same nest at the source site (Salutation Island) were held in larger ‘family’ boxes and released together into the same proto-nest. Between the two translocations of this species to DHI in 2021 and 2022, 118 animals were released at 99 proto-nests, of which 28 were fitted with radio-collars for up to 39 days. Four animals were killed by predators, and four others lost their collars between 10 and 35 days post-release. Dispersal was highly variable, with 13 individuals moving <400 m from their release sites. Ten animals moved more than 2000 m from their release sites (but in proximity to other conspecifics), with three of those moving more than 10,000 m, although one then returned to within 1000 m of the release area. All but 1 of these 10 individuals survived until collar removal (26–39 days post-release). Only two individuals exhibited hyperdispersal, becoming isolated from the rest of the release cohort, and were thus unlikely to contribute to population establishment [89]. Camera traps deployed on a sub-set of proto-nests showed that greater stick-nest rats were using them, not just as initial refuges but also as functional nests with mating behaviour observed. Juveniles were also recorded at proto-nests, confirming that breeding had occurred. These monitoring data were invaluable for assessing progress towards achieving success criteria for the translocation. Overall, the provision of proto-nests was considered successful at promoting initial survival and site fidelity. However, the occupancy of proto-nests was not maintained after the initial release period, despite the ongoing presence of greater stick-nest rats in the release areas (as determined by trapping, scat, and track and camera surveys). It is possible that, despite the augmentation of some nests, when temperatures began to increase in spring, artificial refuges were no longer suitable.

As the only captive-bred species to be released on DHI, dibblers presented a suitable candidate for a soft release, especially after the decision to suspend the radio tracking of released individuals (see above). Furthermore, dibblers had proved very difficult to monitor effectively, with just six detections on a 300 m grid of 25 lured camera traps in over 24 months, and two individuals were successfully trapped in the same timeframe (one was a female, which successfully produced young) [24,25]. At Perth Zoo, dibblers are housed in small glass terraria (1200 mm long × 350 mm wide × 450 mm high [92]) where they are provided with food, refuge and enrichment. It was decided to build pens in the release area that featured natural vegetation but with the provision of food, water and a nest box. During the October 2021 translocation, a trial soft release was conducted with nine dibblers held in individual 2.4 m × 2.4 m pens. Each pen was ~1 m high and consisted of white acrylic plastic panels, which were too smooth for dibblers to climb and constructed around a large, spreading *Acacia ligulata*, *Exocarpos aphyllus* or *Scaevola crassifolia* shrub to provide shelter. Each of these plant species is common in the release area and has abundant accumulations of leaf litter that is understood to hold a high biomass of invertebrate prey [93]. *Scaevola crassifolia* is also common on Boullanger and Escape Islands, where original founders were sourced. Dibblers were held in the pens for 10 nights and were supplied with a similar diet to that at Perth Zoo. Food and water were refreshed daily, and a camera trap was used to monitor activity at the food and water dishes. Dibblers were recaptured regularly to monitor weight and condition, with thresholds for weight loss set, at which the trial would be abandoned. If animals lost more than 10% of their release weight after five days, they were reweighed three days later, and if additional weight loss was more than 5% or over 20% overall, the animal would be released from its pen. After 10 nights, the pen walls were removed, but sites continued to be monitored with camera traps. The 2021 trial was considered successful, with all nine animals surviving until release, and after some initial weight loss, all animals subsequently maintained or gained weight. The main concern from the trial was the presence of grey butcherbirds, a potential predator of dibblers. While dibblers were easily able to evade the butcherbirds, which seemed to be initially attracted by the water supply, this was nevertheless a welfare concern. In 2022, white bird netting was secured over the pen, which was successful in excluding butcherbirds. The 2022 release also involved 2 dibblers per pen (allowing 18 to be released this way) but only same-sex litter mates. A similar result to the trial was recorded, with no apparent welfare issues associated with the release of pairs noted, and observed weight loss was less than in 2021.

We also trialled the use of artificial refuges for dibblers in the form of nest boxes. We released nine dibblers directly from nest-boxes under thick, spreading shrubs with abundant leaf litter but left the nest-boxes in situ. Activity was again monitored using camera traps. By September 2022, dibblers had been recorded at all nest box sites between 66 and 344 days post-release (mean = 237 days). By comparison, dibblers were recorded at release pen sites 5 and 334 days post-release (mean = 165 days). Both represented an improvement on the lured camera grid. Therefore, while the soft release pens were considered successful, the extra allocation of time and resources, when compared to nest box releases, apparently did not confer any advantage in terms of monitoring efficacy.

## 6. Stress and Animal Welfare

Translocations of wildlife are inherently stressful [79]. Acute stress may be precipitated as an adaptive response to the capture, handling, transport or release processes during translocation, while chronic stress may be induced by the consecutive or persistent exposure to these and other stressors [94]. The impact of stress on translocation success rates is not well understood, However, resulting physiological alterations can enhance susceptibility to factors such as predation, disease or dispersal, suggesting that stress may be a critical factor [95].

Stress management was a particularly important consideration for the translocation of rufous hare-wallabies to DHI. Like some other macropodid marsupials [96,97], this species is understood to be vulnerable to stress (or capture) myopathy (exertional rhabdomyolysis) [98,99]. Stress myopathy is frequently associated with the capture and restraint of wild animals, including during translocations, and is often fatal [100]. Rufous hare-wallabies also exhibit other symptoms of stress such as hypersalivation and excessive urination.

A trial translocation of hare-wallabies in August 2017 aimed to refine transport, release and monitoring protocols for these species. To reduce the risk of stress myopathy, several measures were enacted to minimise both the number of external stressors and the physiological response to these stressors. This included prophylactic treatment with antioxidants (selenium and vitamin E (0.2 mL/kg intramuscular (IM)); sedation with diazepam (1.0 mg/kg (IM)); and sedation with azaperone (2 mg/kg (IM)) to maintain animals in a calm state during transport and handling. During capture (using hand nets) chases were limited to <100 m. Handling and noise was minimised, and animals were housed in a cool, dark environment prior to release [94,100]. Transport in 2017 was by boat, which took c.5 h in rough sea conditions to reach DHI from the source sites at Bernier and Dorre Islands. Between capture at source and processing on DHI (~12 h) the 12 rufous hare-wallabies exhibited a mean weight loss of 13% of their total body weight (up to 18%). Banded hare-wallabies only lost ~4% on average in the same period. Weight losses for both species were statistically significant [26]. Animals were located daily by radio tracking for up to 13 weeks post-release. During this period, one mortality was recorded: a male rufous hare-wallaby that was found dead four days post-release. Subsequent necropsy and histopathology confirmed the likely cause as stress myopathy. No other mortalities were recorded during the trial at least nine months post-release (May 2018), but even by this stage, rufous hare-wallabies had failed to recover their original release weight. However, the one recorded incident of stress myopathy, and the significant weight loss, precipitated a change to the protocol for the first full-scale translocation in September 2018, with all subsequent translocations transporting animals by air rather than sea.

The transportation method and the time spent in transit are predicted to be important sources of stress for animals during translocation [94,101]. To reduce the cumulative stress from transit time, in 2018, a helicopter was used in preference to a vessel, reducing travel time to <30 min. However, the noise produced by the helicopter was considered a major stressor, and sedation was still deemed necessary for individuals displaying significant behavioural stress responses. Since the weight changes in 2017 were largely attributed to fluid loss, approximately half the 140 hare-wallabies (90 banded: 50 rufous) were treated with atropine (0.04–0.05 mg/kg subcutaneous (SQ)) for its effects on reducing urination volumes and salivation. Again there were significant weight losses for both species, but for rufous hare-wallabies, the mean loss was just 3%, while for banded hare-wallabies, the mean was 5% and comparable to that of 2017 [87]. There was no significant difference between those animals treated with atropine and controls. No mortalities were recorded in the 12 radio-collared individuals of each species for at least nine months post-release. However, despite the high survival rate, both species showed net weight loss over the monitoring period, indicating that this may relate to seasonal patterns (i.e., natural weight loss during the warmer months between November and April) rather than a delayed effect of the translocation. Given the proximity and the ecological similarity of DHI to Bernier and Dorre Islands, we are confident that this is not an indication of it being a sub-optimal habitat for these species. Since 2017, both species have established well and expanded their extent of occurrence [24].

Effectively managing stress in translocations can be difficult due to the challenging logistics of transporting animals in a timely manner, whilst ensuring other stressors such as noise and handling are also minimised. The use of sedatives to assist in the management of stress myopathy in rufous hare-wallabies was somewhat effective, with just one recorded mortality out of 24 radio-collared individuals. However, weight loss observed in 2017 was indicative that provisions for animal welfare could be improved. Fortunately, this was significantly improved by reducing the time spent in transit, highlighting how important this aspect may be for managing stress in some species.

The project team also implemented measures to reduce stress during the translocation of greater stick-nest rats from the Franklin Islands in South Australia to DHI in 2022, which was a reinforcement of an initial release in 2021 from Salutation Island. Salutation Island is just 100 km from DHI (Figure 1), and, hence, the translocation entailed a much shorter travel duration of ~30 min. In contrast, the journey from the Franklins to DHI is ~2200 km (Figure 1). The project contracted the fastest aircraft that the budget would allow to minimise time in transit. The journey was completed in approximately nine hours, with no mortalities occurring prior to release. The Salutation population had relatively poor genetic diversity compared to the natural population on the Franklins [77], but a PVA was used to estimate the number of Franklins vs. Salutation individuals required to provide an optimal outcome, including reducing the number of animals requiring the nine hour journey [76]. Thus, demonstrating that proximal sources of lower genetic value can be incorporated into the overall strategy to improve welfare outcomes.

To better understand the role of stress in the outcomes of translocations, it would be useful to obtain baselines of, for example, faecal glucocorticoid metabolites (FGM) in source populations to compare with translocated animals. This has been conducted on greater stick-nest rats, finding that changes in FGM were inconsistent (K. Williams-Kelly et al. *unpublished data*): two wild-caught populations showed better acclimation translocation than that of a captive-bred population, highlighting the importance of considering where to source populations to minimise translocation stress. Longitudinal studies may assist with assessing whether translocated populations are chronically stressed [102], but causes of chronic stress can be mitigated in the first instance by ensuring every practical effort is made to reduce the frequency, duration and intensity of manageable stressors and ensure that the translocation is appropriate in the first place.

## 7. Management of Disease Risk

Disease is frequently identified as a risk factor in translocations [2,46,47,103,104], but identifying specific disease hazards and devising practical management strategies for individual taxa and populations can be challenging as it depends on the availability of empirical data for species that may not be well studied. To address the potential uncertainty around disease hazards and the risk they may pose, disease risk analysis (DRA) has become an increasingly popular and valuable tool [105]. DRAs identify, prioritise and assess relevant disease hazards for species’ translocations and, in considering these assessments, then provide practical recommendations for disease risk mitigation.

A good example that demonstrates the value of DRAs is the analysis undertaken for the Shark Bay bandicoot [106,107] (known as the ‘western barred bandicoot’ prior to a taxonomic change [108]), a species for which several key infectious hazards were already known, such as bandicoot papillomatosis carcinomatosis virus type 1 (BPCV1) [109,110] and *Chlamydia* spp. [111]. Recommendations from this DRA were followed during the translocations of this species to DHI, such as developing and implementing monitoring and biosecurity protocols, using strategies to reduce stress during and after translocation (including release site choice and time of year; see Section 5) and post-translocation monitoring (e.g., live-capture trapping). This included rejecting any bandicoots displaying potential symptoms of BPCV1 or *Chlamydia* spp. infection. Unfortunately, one bandicoot that had already arrived on DHI was noted to have a suspicious lesion on its foot, with a gross appearance consistent with possible BPCV1 infection. Circumstances meant the individual could not be returned to its origin (Bernier Island), so the decision was made to euthanise it [88]. Swabs taken later tested negative for BPCV1. However, another individual with a similar lesion that circumstances allowed to be returned to Bernier Island, subsequently tested positive for BPCV1. Such a conservative approach to biosecurity is necessary to avoid the inadvertent introduction of a potentially significant pathogen to the DHI translocated population, and, to date, no bandicoots caught on DHI have tested positive for BPCV1 despite ongoing monitoring.

To facilitate the development of DRAs for other translocations, a Veterinary Resident position was established via a partnership between Murdoch University and the DBCA. The resident was tasked with undertaking a combined DRA for all rodents that were either planned for translocation or extant on the island (seven species total) and single-species DRAs for both the boodie and chuditch. The latter are still in progress. DRAs were also undertaken for dibblers [112] and western grasswrens [113] as student projects. Recommendations from these DRAs have initiated new projects to fill identified knowledge gaps, such as baseline health examinations of rodents on DHI and source populations (F. Knox et al. *unpublished data*) and a project identifying and describing mites (*Demodex* spp.) in dibblers (C. Bowry et al. *unpublished data* [114]).

Quarantining animals during translocations is a potentially critical step to mitigate disease risks [115,116]. However, the need to quarantine animals prior to release should be assessed on a case-by-case basis [2] as confining animals to captivity may impose additional stress and welfare implications and may even increase disease risks for an otherwise healthy population [104,117,118]. Quarantine may be more appropriate for captive-bred founders rather than those being translocated wild-to-wild [119]. The DRA for the Shark Bay bandicoot translocation included quarantine as potentially desirable in the translocation pathway, but logistical and welfare issues associated with attempting to maintain the entire founder cohort in captivity were identified [107]. These constraints were broadly applicable to all species, except for the dibbler, which undergo quarantine and disease screening on entering captivity at Perth Zoo and are screened again prior to release [52].

Another important aspect of managing disease in translocations is pre- and post-release health surveillance [2,104]. All DRAs that have been used to inform translocation strategies to DHI have made recommendations for the collection of baseline health data prior to release as well as ongoing post-release health surveillance. This includes ongoing health assessments in the field and routine necropsy of wildlife deaths incorporating histopathology. The implementation of post-release surveillance and baseline health data collection not only enables an early warning system, so that disease issues are detected in a timely fashion, but also provides useful information to further refine DRAs for the benefit of future conservation translocations.

## 8. Genetic Management

The maintenance of genetic diversity has been understood to be vitally important for biodiversity conservation for some time [120,121]. Genetic considerations must be made at multiple stages within the translocation process by the selection of appropriate source populations, determination of founder group sizes to capture representative genetic variation and ongoing monitoring and management to ensure its persistence [122]. Nowhere is the need to consider genetics greater than in conservation translocations [123].

Genetics studies have informed planning for the DHI fauna reconstruction, in terms of preferred source populations and desired founder sizes [10]. Information on some species was obtained from previous genetic studies (rufous hare-wallaby [124,125], greater stick-nest rat [77] and Shark Bay bandicoot [126]) or from studies initiated either in collaboration with or conducted by the DHI project (banded hare-wallaby [73], dibbler [57], Shark Bay mouse [75] and western grasswren [127]). In the case of the western grasswren, no previous population genetics studies had been undertaken for this species, and understanding whether there was any sub-structuring within the proposed source population on the Shark Bay mainland was important for determining where and in what proportion founders should be sourced. This study found that there was evidence of substantial differentiation between the Peron Peninsula and Hamelin Station subpopulations (Figure 1), and it was recommended that founders be sourced from both locations to ensure the establishment of a genetically representative population on DHI [127].

Harvesting animals for translocation can impact the genetic diversity of the source population and form the basis of the genetic diversity of the translocated population. In conservation translocations, tension arises between choosing sufficiently large numbers of individuals to establish genetically diverse founding populations whilst ensuring minimal demographic impact to the source. As discussed in Section 4, genetically informed PVAs have been invaluable in selecting optimal translocation strategies for DHI that aim to (a) maximise genetic diversity and the likelihood of the successful establishment of the translocated population and (b) reduce the impact to the demographic stability and genetic diversity of the source population(s). Genetic PVAs are run for multiple generations (typically 50 to 100 years) to ensure that optimal strategies result in the long-term maintenance of genetic diversity in both translocated and source populations. Genetic data have been incorporated into PVAs to inform translocation strategies for all but one DHI species, the rufous hare-wallaby, although a PVA has subsequently been developed to serve as a predictive model for monitoring the long-term maintenance of genetic diversity (K. Ottewell et al. *unpublished*).

The strategic framework for the DHI fauna reconstruction incorporated time-bound genetic monitoring and identified potential issues that needed to be investigated either prior to or after the translocations took place [10]. The framework also identified the potential need for additional reinforcement translocations to augment populations if the genetic diversity of the founder cohort had not been sufficiently retained or if inbreeding had been identified as a potential problem. As such, tissue has been collected from all translocated founders and new island-born progeny (captured via monitoring programs) for subsequent genetic ‘audits’ to continue to assess each population.

## 9. Summary

Translocation is, and will remain, an important and popular conservation tool for a variety of purposes, and it will continue to grow in popularity [5]. Therefore, it is important that individual projects communicate their findings so that others may learn and adapt their methods. Multispecies translocation programs are particularly challenging, and the *Return to 1616* fauna reconstruction represents a complex case study. In this review, we have highlighted many of the challenges this project has faced and how we have addressed and adapted to these challenges. We discuss many key aspects of the translocation program, including modelling to support decision making, disease risk and animal welfare considerations, genetic management, and release strategies, mitigating impacts on source populations and improving the effectiveness of post-release monitoring. While many of the approaches we discuss are not strictly ‘novel’, we have applied and combined these approaches in innovative ways to resolve important questions and address particular challenges, in many cases resulting in an improved translocation outcome.

We acknowledge that this review is not exhaustive in terms of all aspects of translocations that need to be considered by practitioners. For example, we do not address the suitability of habitat at release sites, which has been identified as a key reason for translocation failure elsewhere [81]. However, since the removal of sheep and goats, the recovery of vegetation on DHI has been monitored using remote sensing and indicates that a large proportion of the island’s vegetation cover has increased significantly [20]. Release sites were also selected based on the similarities to source population locations.

Another important component is ongoing monitoring, which is crucial for evaluating the long-term success of individual translocations and the broader faunal reconstruction program [82,83] and is most commonly reported as causing difficulties for translocation practitioners [9]. The long-term monitoring program for the DHI fauna reintroductions is still being developed, and we are not yet able to report on its success or other outcomes. Nevertheless, some exciting innovations in this area have already been made, such as the use of DNA extracted from faecal pellets (faecal DNA) to identify individual banded hare-wallabies [128], data which can then derive population estimates using a spatially explicit capture–recapture (SECR) framework without having to capture individual animals (e.g., as has been achieved for greater bilbies (*Macrotis lagotis*) [129]).

Throughout the project, we have identified numerous knowledge gaps, which require further research. To address some of the main research priorities, collaborations with universities have been crucial, particularly via student projects. For example, the Veterinary Resident (discussed in Section 7) has supported the project with not only the development of DRAs but also the initiation of further research to address identified knowledge gaps. The uncertainty around the presumed conservation introduction of banded and rufous hare-wallabies to DHI was the motivation to establish a student project investigating the diet of these species, using environmental DNA extracted from scats to identify food plants at both source and recipient sites (R. Stover *unpublished data*). Avenues for future research include how the reintroduction of locally extinct fauna may be contributing to the ecological restoration of DHI ecological processes, especially the effect of digging mammals such as Shark Bay bandicoots and the two bettong species.

Based on our experiences with the *Return to 1616* fauna reconstruction, we can make the following suggestions for others undertaking similarly complex translocation programs to consider:-Employ the use of decision support tools, including PVAs and DRAs, to inform translocation strategies and identify the most important risks that may need to be mitigated.-Trial novel and innovative techniques, particularly if there is a likelihood of improved translocation and animal welfare outcomes, ideally within an experimental framework to fully assess their efficacy.-Proactively communicate findings so that other projects can benefit from the collective learnings of the translocation community, including reasons for failures as well as successes, as these will be of value to those seeking to avoid similar pitfalls. The *Return to 1616* project has greatly benefited from the publication of results from other translocations elsewhere.

Translocation is a valuable tool in restoring biodiversity values, but it often requires large allocations of resources, can have major animal welfare implications and is prone to failure. However, as this review demonstrates, by the thoughtful use of decision-support tools and promoting novel and innovative thinking, a project as large as the *Return to 1616* fauna reconstruction can overcome many of the challenges of large multispecies translocation programs.

## Figures and Tables

**Figure 1 animals-13-02762-f001:**
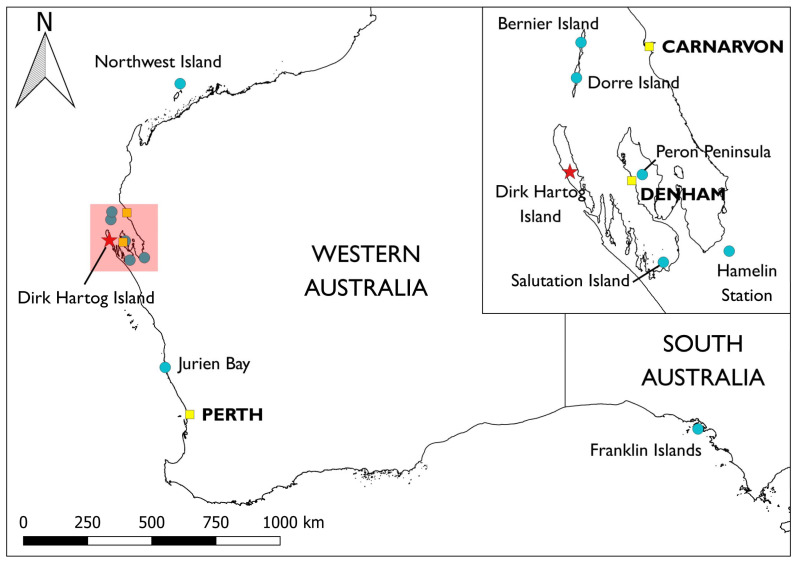
Western and South Australia showing locations of Dirk Hartog Island (red star), source populations (blue circles) and cities/towns (yellow squares). Inset shows Shark Bay area.

**Table 1 animals-13-02762-t001:** Summary of species planned for translocation to Dirk Hartog Island (listing refers to conservation status under the Environmental Protection and Biodiversity Conservation (EPBC) Act 1999 (V, vulnerable; E, endangered; N, not listed). PVA, Population viability analysis; DRA, disease risk analysis). ‘Released’ refers to number of individuals released. ‘Source(s)’ refers to the locations where founders were originally sourced. Details of releases for final five species are still to be confirmed.

Common Name	Scientific Name	Listing	Years	Released	Source(s)	PVA	DRA
Banded hare-wallaby *	*Lagostrophus fasciatus*	V	2017–18	102 ^H^	Bernier Is.; Dorre Is.	Y	N
Rufous hare-wallaby *	*Lagorchestes hirsutus*	V	2017–19	112 ^H^	Bernier Is.; Dorre Is.	N	N
Shark Bay bandicoot	*Perameles bougainville*	E	2019–20	99 ^H^	Bernier Is.; Dorre Is.	Y	Y
Dibbler	*Parantechinus apicalis*	E	2019–	137 ‡,^M^	(Boullanger Is.; Escape Is.; Whitlock Is.) via Perth Zoo	Y	Y
Shark Bay mouse	*Pseudomys gouldii*	V	2021–22	130 ^A^	Bernier Is.; Northwest Is.	Y	Y
Greater stick-nest rat	*Leporillus conditor*	V	2021–22	122 ^A^	East Franklin Is.; West Franklin Is.; Salutation Is.	Y	Y
Western grasswren †	*Amytornis textilis*	N	2022	85 ^H^	Hamelin Station; Peron Peninsula	Y	Y
Brush-tailed mulgara	*Dasycercus blythi*	N	2023	100 ^M^	Matuwa Kurrara Kurrara	Y	N §
Desert Mouse	*Pseudomys desertor*	N	-	-	-	-	Y
Heath Mouse	*Pseudomys shortridgei*	E	-	-	-	-	Y
Woylie	*Bettongia penicillata*	E	-	-	-	-	-
Boodie	*Bettongia lesueur*	V	-	-	-	-	-
Chuditch	*Dasyurus geoffroii*	V	-	-	-	-	-

* Treated as a conservation introduction; † subspecific substitution for extinct endemic taxon; ‡ more releases planned; § DRA for dibbler considered to be applicable given lack of available studies on disease in dasyurids; ^H^ hard release only; ^A^ hard release with some provision of artificial refuges; ^M^ mixed soft and hard release with provision of artificial refuges.

## Data Availability

The data presented in this study are available on request from the corresponding author.

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
