# Peer review of "Return to 1616: Multispecies Fauna Reconstruction Requires Thinking Outside the Box"

_animals, 2023, doi:10.3390/ani13172762_

Round 1

Reviewer 1 Report

Your paper is well written, clearly structured and easy to read.

There are always some 'buts', mostly minor, and hopefully they will help ensure your contribution gains a wide audience. Most of my comments relate to the Introduction which I would like to see expanded to help 'set the scene'. If you can, please add a few sentences where appropriate to cover the following:

How well was the original fauna described, was there a good description of botanical richness and habitat diversity? Briefly, how did domestic animal grazing effect the structure of natural habitats

Line 60: island size - please give approximate length/breadth in km (80x10) as it is hard to extract this from the size given in hectares or from Figure 1.

Line 66: The term 'weeds' needs some explanation, are some of them native (that thrived in the presence of sheep and goats) or are they mostly non-native/introduced?

Table 1: either here or in the text, please give the dietary guild of the re-introduced species for those readers that are unfamiliar with Australian mammals.

Line 184: probably replace 'this year' with 2023.

Line 423: reference 87 - is this an unpublished thesis?

Line 435: your discussion of the potential impact of butcherbirds raises a question that may have not been covered so far - the presence and diversity of of (small mammal) predators. Perhaps slot this in in the Introduction?

Line 479: 'Losses for both species' suggests mortality -  maybe insert 'Weight'.

Line 485: failure to regain weight post-release -does this suggest some aspect of the habitat is 'not right'?

Line 501: insert '12' before 'radio-collared'.

Line 631: typo 'genetic'

Before Summary/line 647: What about inserting a Table/series of bullet poits or even a short paragraph on 'Knowledge Gaps'? You mention this in line 577. One topic could be understanding seasonal diet variation and its linkage to body mass changes.

Author Response

Your paper is well written, clearly structured and easy to read.

  • Thank you and we are grateful to you for taking the time to provide your constructive review in such a timely manner.

There are always some 'buts', mostly minor, and hopefully they will help ensure your contribution gains a wide audience. Most of my comments relate to the Introduction which I would like to see expanded to help 'set the scene'. If you can, please add a few sentences where appropriate to cover the following:

How well was the original fauna described, was there a good description of botanical richness and habitat diversity? Briefly, how did domestic animal grazing effect the structure of natural habitats

  • The original fauna is largely known through subfossil remains, discussed in lines 99-106 of the original manuscript. This is also described in the supplementary table.
  • Flora collections were made by many of the early explorers but there are no accounts that allow us to assess how the vegetation has changed. The best way to evaluate this impact is through the post-eradication recovery which has been carefully monitored using remote sensing. We have added the following text to this effect: ‘Sheep and goats caused considerable damage to the island’s vegetation, especially around artificial sources of water. The overall impact of ungulates on the island’s vegetation can best be understood through the rapid and significant recovery since the eradications [20].’

Line 60: island size - please give approximate length/breadth in km (80x10) as it is hard to extract this from the size given in hectares or from Figure 1.

  • Included text: (~80km long and 3-11km wide)

Line 66: The term 'weeds' needs some explanation, are some of them native (that thrived in the presence of sheep and goats) or are they mostly non-native/introduced?

  • Replaced ‘weeds’ with ‘non-native plants’

Table 1: either here or in the text, please give the dietary guild of the re-introduced species for those readers that are unfamiliar with Australian mammals.

  • This has been included in supplementary table.

Line 184: probably replace 'this year' with 2023.

  • Changed ‘this year’s’ to ‘the 2023’

Line 423: reference 87 - is this an unpublished thesis?

  • Yes and reference has been corrected to state this

Line 435: your discussion of the potential impact of butcherbirds raises a question that may have not been covered so far - the presence and diversity of of (small mammal) predators. Perhaps slot this in in the Introduction?

  • We are concerned about adding significantly to the length of this manuscript but we have added a paragraph to address this: ‘The apex terrestrial predator on DHI was formerly the chuditch, but in its absence there are other predators of small mammals and birds present, such as sand monitor (Varanus gouldii), mulga snake (Pseudechis australis), gwardar (or western brown snake) (Pseudonaja mengdeni), Children’s python (Antaresia childreni) and grey butcherbird (Cracticus torquatus). The two largest predators on the island are the wedge-tailed eagle (Aquila audax) and the white-bellied sea-eagle (Haliaeetus leucogaster), the former an apparently scarce visitor to DHI. All these species are natural predators of the former fauna assemblage of the island.’

Line 479: 'Losses for both species' suggests mortality -  maybe insert 'Weight'.

  • Inserted ‘weight’ before ‘losses’

Line 485: failure to regain weight post-release -does this suggest some aspect of the habitat is 'not right'?

  • We discuss our hypothesis for this failure to regain weight in lines 505-507, relating to seasonal patterns. Monitoring of hare-wallabies was limited by the battery life of the transmitters, which meant removing all collars in May prior to the collar batteries dying. This meant we monitored the hare-wallabies between September and May, which represents the period of the year with the least rainfall. Therefore, we think it is likely that the failure to regain the weight they had in September (when they have had several months of winter rainfall to build up condition) is due to the period during which they were monitored. Bernier and Dorre Islands, where the animals were sourced, is ostensibly similar to Dirk Hartog Island in vegetation structure but DHI is more floristically diverse and receives on average more rainfall.
  • However, we feel it is important to emphasise this was not due to unsuitable habitat and have included the following: ‘Given the proximity and the ecological similarity of DHI to Bernier and Dorre Islands, we are confident this is not an indication of sub-optimal habitat for these species. Since 2017, both species have established well and expanded their extent of occurrence [24].’

Line 501: insert '12' before 'radio-collared'.

  • Rephrased sentence to read ‘No mortalities were recorded in the 12 radio-collared individuals of each species…’

Line 631: typo 'genetic'

  • Corrected to ‘genetic’

Before Summary/line 647: What about inserting a Table/series of bullet poits or even a short paragraph on 'Knowledge Gaps'? You mention this in line 577. One topic could be understanding seasonal diet variation and its linkage to body mass changes.

  • Throughout the project, we have identified numerous knowledge gaps which require further research. To address some of the main research priorities, collaborations with universities have been crucial, particularly through student projects. For example, the Veterinary Resident (discussed in Section 7) has not only supported the project through the development of DRAs, but also the initiation of further research to address identified knowledge gaps. The uncertainty around the presumed conservation introduction of banded and rufous hare-wallabies to DHI was the motivation to establish a student project investigating the diet of these species, using environmental DNA extracted from scats to identify food plants at both source and recipient sites (R. Stover unpublished data). Avenues for future research include how the reintroduction of locally extinct fauna may contribute to the restoration of ecological processes, especially the effect of digging mammals such as Shark Bay bandicoots and the two bettong species.

Reviewer 2 Report

Dear authors,

Please make additions to the manuscript and, if possible, shorten the text. The article carries a great semantic load, which is not always justified by the title of the manuscript.

Author Response

We thank the reviewer for their constructive feedback and for taking the time to provide a review within such a short time-frame.

We thank the reviewer for making suggestions of where we can reduce the text. However, other reviewers have requested additional information, which has made reducing the overall length of the document challenging.

Line 50: Add: In many countries, there is a historical increase in the proportion of alien species of animals and their gradual displacement of native species (Darvish et al. 2014, Andreychev and Kuznetsov 2020). However, in Australia this problem is particularly acute.

  • We have now included a statement to address this point: ‘Invasive predators are a key driver of global biodiversity loss, not least in Australia [6] and mitigating this is an important motivation for undertaking translocations [1]. In Australia, translocations of threatened fauna to ‘safe havens’ (i.e. fenced reserves or islands, free from non-native predators) have been highly effective management actions [7].

Line 58: What is the purpose of this information within this manuscript?

  • Text removed.

Line 62: All of these species must be listed.

  • Included additional supplementary table which captures information about all species, extant or locally extinct.

Line 63: Justification is needed for the settlement of the territory for each of these species. This is especially true for those two out of thirteen species that might not have been present in the extinct fauna earlier.

  • We think that reviewer is referring to how release sites for each species were determined and justified in terms of the inter-specific interactions. We feel this is addressed in the ensemble modelling exercise discussed in Section 2, where we considered the location of release sites and how that might affect the outcomes of each translocation, i.e. through temporal and spatial separation.

Line 80-81: What are your suggestions for these species?

  • Laughing doves are not considered an invasive species in Australia and we do not anticipate that this species will present a specific threat to any extant or translocated fauna.
  • House mice are an invasive species and may compete with native species. However, an eradication of house mice on an island the size of Dirk Hartog would be a Herculean task and would likely have huge detrimental impact on the two native species of rodent.
  • House mice abundance will be monitored through the small vertebrate monitoring in Section 2.
  • We have included the additional text: ‘Laughing doves are not considered to be an invasive species in Australia and their presence was not considered to be a threat to the extant or translocated fauna. House mice are an invasive species that may outcompete native species [22] but, based on estimates for another WA island [23], the eradication of house mice from an area the size of DHI would cost more than US$47 million. Furthermore, any eradication program would likely endanger the extant (and translocated) small vertebrates on the island. The population dynamics of house mice are monitored annually through trapping surveys (see Section 2). Monitoring indicates that native rodents are more abundant than house mice [24], which are not currently considered to be a problem.’

Line 97: It is not necessary

  • We feel it is important for context to provide a time-frame for this project as it dictates what is possible and have retained this statement.

Lines 256-273: These are the results for another article

  • Respectfully, we would prefer to include these results. This was a major issue for the post-release monitoring of dibblers that we believe was directly due to having been sourced from captivity. We have included an additional sentence underlining the context of these results:

‘However, this highlights an unforeseen issue with sourcing founders from captivity that other programs may need to consider’.

Lines 571-580: The text of the manuscript contains a lot of additional information, due to which it is overloaded with semantic load. This information does not always fit within the title of the manuscript. The text needs to be shortened.

  • Our intent was to bring the reader along on our journey of discovery as the translocations progressed – dealing with the various challenges along the way – with the intent to impart some of our learnings. For example, in the case highlighted by this reviewer we discuss how we were able to initiate comprehensive Disease Risk Analyses for so many species within the constraints of the project’s resources. We also highlight the added value we obtained from the DRA process. As such, we would prefer to keep this relatively short passage, particularly as it provides additional context for the rest of this section.

Reviewer 3 Report

The manuscript entitled „Return to 1616: multispecies fauna reconstruction requires thinking outside the box” by Saul Cowen and colleagues, submitted to the Animals magazine, describes problems of conservation translocations, and presents some new data on the project in the Dirk Hartog Island in Western Australia. It is rather descriptive manuscript; I think it is interesting and well written summary on problems with translocation actions. Information on such programs – on failures and successes (see for example lines 685-689) – are crucial for many scientists. Thus, this manuscript is worth to be published, however, I have several comments, which could help to improve it.

I am not sure, if in the title, the sentence “requires thinking outside the box” is the best one. I agree that (line 28-29) “These multispecies translocations are complex and need to consider the requirements of each individual species as well as the influence of likely interactions among them.”, however, Authors have written that they present (lines 32-33): “a synthesis of the innovative approaches undertaken in restoring the former faunal assemblage of Dirk Hartog Island and the key learnings”.
Please forgive me for my possible misunderstanding, but what exactly are the “innovative approaches”? I strongly agree that – for example – the release strategies are important for such programs, I appreciate that the problem of stress and animal welfare is considered and discussed in the manuscript, as well as using the Population Viability Analysis, etc. But I am not sure what Authors understand as the “innovative approaches”.
In the summery Authors have written that they (lines 655-658) “discuss many key aspects of the translocation program, including modelling to support decision-making, disease risk and animal welfare considerations, genetic management, release strategies, mitigating impacts on source populations and improving the effectiveness of post-release monitoring”.
I strongly believe it is correct way for translocation actions, but – for me – it is not “innovative approaches”. It should be stated – especially in the Abstract and the Summary sections – what are the “innovative approaches” in the study.

Several comments
The ‘Return to 1616 plan’ concerns on 12 mammal species. But, what about other groups of animals, especially other vertebrates? There is some information in the subject only (see line 165 and the next ones), however, more information – why the plan concerns on mammals only, how it could influence on, e.g., reptiles, etc. – is recommended.
Additionally, I think the sentence “Return to 1616 fauna reconstruction” could be replaced by – for example – “Return to 1616 mammal fauna reconstruction”, especially in the summary section. 

line 48 “...varies considerably, [3,5,7] and...” ­– the comma should be after ‘[3,5,7]’, I think.

Table 1. Summary of translocations undertaken…:
Released” – Number of released individuals?
‘Source(s)’ – Source(s) populations?

lines 130-131 “the DHI fauna reconstruction seeks to establish a functioning ecosystem" – I think that the sentence should be changed: if now there is no functioning ecosystem in the area (before finish of the translocation action)?

line 498 “Again there were significant weight losses (p < 0.001)”

This manuscript is rather descriptive paper, thus, I am not sure, if the “p” values is necessary here. However, if you think, that such values are important, I suggest to show the statistical results (here, additionally, the sample size, the statistical test value; not only the p value), and add such values in other places, e.g. for lines 500-501 “There was no significant difference between those animals treated with atropine and controls.” [why there is no ‘p’ value, but in the line 498 such value is presented?]

line 606 “[113,114]” is repeated.

Author Response

The manuscript entitled „Return to 1616: multispecies fauna reconstruction requires thinking outside the box” by Saul Cowen and colleagues, submitted to the Animals magazine, describes problems of conservation translocations, and presents some new data on the project in the Dirk Hartog Island in Western Australia. It is rather descriptive manuscript; I think it is interesting and well written summary on problems with translocation actions. Information on such programs – on failures and successes (see for example lines 685-689) – are crucial for many scientists. Thus, this manuscript is worth to be published, however, I have several comments, which could help to improve it.

  • Thank you for the constructive feedback and for taking the time to review our manuscript within the short time-frame.

I am not sure, if in the title, the sentence “requires thinking outside the box” is the best one. I agree that (line 28-29) “These multispecies translocations are complex and need to consider the requirements of each individual species as well as the influence of likely interactions among them.”, however, Authors have written that they present (lines 32-33): “a synthesis of the innovative approaches undertaken in restoring the former faunal assemblage of Dirk Hartog Island and the key learnings”.
Please forgive me for my possible misunderstanding, but what exactly are the “innovative approaches”? I strongly agree that – for example – the release strategies are important for such programs, I appreciate that the problem of stress and animal welfare is considered and discussed in the manuscript, as well as using the Population Viability Analysis, etc. But I am not sure what Authors understand as the “innovative approaches”. In the summery Authors have written that they (lines 655-658) “discuss many key aspects of the translocation program, including modelling to support decision-making, disease risk and animal welfare considerations, genetic management, release strategies, mitigating impacts on source populations and improving the effectiveness of post-release monitoring”.
I strongly believe it is correct way for translocation actions, but – for me – it is not “innovative approaches”. It should be stated – especially in the Abstract and the Summary sections – what are the “innovative approaches” in the study.

  • We appreciate that, for those with a broad knowledge of the conservation translocation literature, some of the approaches we discuss may not be considered ‘novel’, as may have been conceived and implemented elsewhere in the world. We agree that some aspects of what we discuss should be considered ‘best practice’, such as Disease Risk Analyses (although how widespread ‘best practice’ is within Australia or globally is another matter). However, we feel there is an important distinction to be made between approaches that are entirely novel, and those that show innovative use of existing approaches.
  • For example, PVA has been in use in conservation management for some time, but we feel that our use of PVA, by incorporating genetic data and managing risk around translocations is an innovative use of PVA.
  • Some of our innovations demonstrate our adaptive approach, for example, using learnings from another taxon to improve the reproductive output of dibblers in captivity.
  • Moreover, we present some examples which we feel strongly are both novel and innovative, such as the provision of artificial refuges for Shark Bay mice and greater stick-nest rats and the use of complex ensemble modelling that consider interspecific interactions to inform the translocation order.
  • As such, we feel that we have ‘thought outside the box’ in many respects in this project. We hope the reviewer also forgives the deliberate play-on-words in the title. Given the questions around innovation, we have now stated: ‘While many of the approaches we discuss may not be considered strictly ‘novel’, we have applied and combined these approaches in innovative ways to resolve important questions and address particular challenges, in many cases resulting in an improved translocation outcome.’

Several comments
The ‘Return to 1616 plan’ concerns on 12 mammal species. But, what about other groups of animals, especially other vertebrates? There is some information in the subject only (see line 165 and the next ones), however, more information – why the plan concerns on mammals only, how it could influence on, e.g., reptiles, etc. – is recommended.

  • The rationale behind why the listed species were chosen for translocation is discussed in lines 99-106 in the original manuscript but we have now included a supplementary table which provides more information for each species. No reptiles are known to have been extirpated from Dirk Hartog Island, but one additional bird and one additional mammal are thought to have. We have added two lines to clarify this: ‘Two other species are thought to have become extinct on DHI: the rock parrot (Neophema petrophila), which was previously observed on the island but is now apparently extinct [30] and the rakali or water rat (Hydromys chrysogaster) which was speculated to have also occurred [10]. Both species were considered capable of natural recolonisation from nearby populations in Shark Bay. No other taxa, including plants, amphibians, reptiles or invertebrates are known to have been extirpated from DHI.
  • When undertaking the ensemble modelling (lines 145-164), we did consider all small vertebrates, including reptiles, although with over 40 species to consider, we had to assign these to guilds rather than consider individual species.
  • Further more in Section 2, we discuss how we are monitoring the impact of the eradications of goats, sheep and feral cats and the subsequent fauna translocations on the extant fauna (which includes the 40 species of reptile).

Additionally, I think the sentence “Return to 1616 fauna reconstruction” could be replaced by – for example – “Return to 1616 mammal fauna reconstruction”, especially in the summary section. 

  • As discussed, the fauna reconstruction includes one bird species (western grasswren), hence describing it as a fauna reconstruction.

line 48 “...varies considerably, [3,5,7] and...” ­– the comma should be after ‘[3,5,7]’, I think.

  • Agreed, comma moved from before citations to after.

Table 1. Summary of translocations undertaken…:
“Released” – Number of released individuals?
‘Source(s)’ – Source(s) populations?

  • Have included the following text in the table heading: ‘Released’ refers to number of individuals released. ‘Source(s)’ refers to the locations where founders were originally sourced.

lines 130-131 “the DHI fauna reconstruction seeks to establish a functioning ecosystem" – I think that the sentence should be changed: if now there is no functioning ecosystem in the area (before finish of the translocation action)?

  • Agreed, this is somewhat misleading, although we are missing many aspects of a truly functional ecosystem (e.g. fossorial mammals, apex terrestrial predators), the ecosystem itself still ‘functions’. We have addressed this by changing the sentence to: ‘…seeks to restore the island’s ecosystems to a similar condition to their pre-European state…’

line 498 “Again there were significant weight losses (p < 0.001)”

This manuscript is rather descriptive paper, thus, I am not sure, if the “p” values is necessary here. However, if you think, that such values are important, I suggest to show the statistical results (here, additionally, the sample size, the statistical test value; not only the p value), and add such values in other places, e.g. for lines 500-501 “There was no significant difference between those animals treated with atropine and controls.” [why there is no ‘p’ value, but in the line 498 such value is presented?]

  • Agreed, for consistency we have removed this level of information.

line 606 “[113,114]” is repeated.

  • Extra citations have been removed.